# Oxidative Stress Parameters in Goitrogen-Exposed Crested Newt Larvae (*Triturus* spp.): Arrested Metamorphosis

**DOI:** 10.3390/ijerph18189653

**Published:** 2021-09-13

**Authors:** Jelena P. Gavrić, Svetlana G. Despotović, Branka R. Gavrilović, Tijana B. Radovanović, Tamara G. Petrović, Maja Ajduković, Tijana Vučić, Milena Cvijanović, Caterina Faggio, Marko D. Prokić

**Affiliations:** 1Department of Physiology, Institute for Biological Research “Siniša Stanković”, National Institute of Republic of Serbia, University of Belgrade, Bulevar Despota Stefana 142, 11060 Belgrade, Serbia; jelena.gavric@ibiss.bg.ac.rs (J.P.G.); despot@ibiss.bg.ac.rs (S.G.D.); perendija@ibiss.bg.ac.rs (B.R.G.); tijana@ibiss.bg.ac.rs (T.B.R.); tamara.petrovic@ibiss.bg.ac.rs (T.G.P.); marko.prokic@ibiss.bg.ac.rs (M.D.P.); 2Department of Evolutionary Biology, Institute for Biological Research “Siniša Stanković”, National Institute of Republic of Serbia, University of Belgrade, Bulevar Despota Stefana 142, 11060 Belgrade, Serbia; mslijepcevic@ibiss.bg.ac.rs (M.A.); milena.cvijanovic@ibiss.bg.ac.rs (M.C.); 3Faculty of Biology, Institute of Zoology, University of Belgrade, Studentski trg 16, 11000 Belgrade, Serbia; tijana.vucic@bio.bg.ac.rs; 4Department of Chemical, Biological, Pharmaceutical and Environmental Sciences, University of Messina, Viale Ferdinando Stagno d’Alcontres, Santa Agata, 3198166 Messina, Italy

**Keywords:** thiourea, thyroid hormones, metamorphosis, antioxidative compounds, amphibians

## Abstract

Thiourea is an established disruptor of thyroid hormone synthesis and is frequently used as an inhibitor of metamorphosis. The changes caused by thiourea can affect processes associated with the oxidative status of individuals (metabolic rate, the HPI axis, antioxidant system). We investigated the parameters of oxidative stress in crested newt (*Triturus* spp.) larvae during normal development in late larval stage 62 and newly metamorphosed individuals, and during thiourea-stimulated metamorphosis arrest in individuals exposed to low (0.05%) and high (0.1%) concentrations of thiourea. Both groups of crested newts exposed to thiourea retained their larval characteristics until the end of the experiment. The low activities of antioxidant enzymes and the high lipid peroxidation level pointed to increased oxidative stress in larvae at the beginning of stage 62 as compared to fully metamorphosed individuals. The activities of catalase (CAT) and glutathione-S-transferase (GST) and the concentration of sulfhydryl (SH) groups were significantly lower in larvae reared in aqueous solutions containing thiourea than in newly metamorphosed individuals. The high thiourea concentration (0.1%) affected the antioxidative parameters to the extent that oxidative damage could not be avoided, contrary to a lower concentration. Our results provide a first insight into the physiological adaptations of crested newts during normal development and simulated metamorphosis arrest.

## 1. Introduction

Although many groups of animals occupy distinct niches as larvae and adults and experience radical ontogenetic changes, amphibian metamorphosis can be considered the most dramatic of all [1,2]. Metamorphosis, as a central amphibian life-history trait, is dependent on the interplay of body growth, developmental progression, and environmental conditions, and the whole process is integrated and controlled by the neuroendocrine system. In terms of the severity of changes during metamorphosis, tailed amphibians (salamanders, urodeles) are between tailless amphibians (anurans), which are characterized by the most drastic alterations, and legless amphibians (caecilian), which undergo the least modification. Remodeling in salamanders includes loss of external gills, closure of gill slits, resorption of tail fins, formation of eyelids and the nasolacrimal duct, as well as loss of Leydig cells in the skin [1,3,4,5].

Hormonal control of amphibian metamorphosis has been thoroughly studied and involves the integration of several hormones such as thyroid hormones (thyroxine T4 and triiodothyronine T3), prolactin, sex steroids, and corticosteroids [6,7,8,9,10,11,12]. Thyroid hormones (THs) are primary morphogens that regulate metamorphosis and stimulate diverse and multiple morphological, physiological, and biochemical responses in larval tissues [13]. Moreover, in their absence, metamorphosis does not occur [14]. As in other vertebrates, the production of THs by the thyroid gland is tightly controlled by the hypothalamic-pituitary-thyroid axis, whereas in target tissues monodeiodinase enzymes convert T4 to the biologically more active T3 [1,13]. Secretion of T3 and T4 increases during larval development and reaches a peak at metamorphic climax in both anurans and urodeles [8,15,16].

Amphibian larvae can be reliable bio-indicator organisms in monitoring aquatic health and examining the effects of various xenobiotics that reach the environment [17]. A tight relation between the neuroendocrine system and development makes amphibians a suitable model for studying the effects of endocrine disruptors, compounds that interfere with the endocrine systems and can cause changes in normal developmental processes [18]. Inhibitors of THs synthesis, also known as goitrogens, prevent metamorphosis, while treatment with exogenous THs accelerates this process [19,20]. Thiourea is a well-known goitrogenic sulfur-containing compound that blocks iodine organification and thus stops the synthesis of THs [21]. It has a wide range of applications in medicine, the electronic industry, agriculture, and the preparation of heterocyclic and plastic materials [22,23]. Studies have shown that thiourea is an endocrine-disrupting chemical that can alter the thyroid function of different organisms (fish, amphibians, lizards, mammals) [20,24,25]. It was shown that thiourea acts via inhibition of thyroid peroxidase, an enzyme that is indispensable to THs synthesis [26,27]. The endocrine-modulating properties of thiourea were evidenced by depression of the levels of THs, a lower metabolic rate, as well as other symptoms of hypothyroidism. In thiourea-treated fish, the level of T4 was unchanged; however, T3 was significantly lower than in the control group, which pointed to the effect of thiourea on the rate of T4 deiodination [28]. The goitrogenic action of thiourea and its effect on amphibian development has been investigated in several studies [20,29,30]. Larvae exposed to this antithyroid agent continue to grow at the same or even faster rate than the control group, but the morphology, internal histology, and biochemistry remain larval with little, if any, indication of progress toward metamorphosis [14]. Thiourea treatment during the late larval stage of *Triturus* newts provoked hypertrophy and hyperplasia of thyroid follicles, causing exhaustion of the thyroid gland and inhibition of hormone synthesis [31]. The presence of thiourea kept Siberian salamanders (*Salamandrella keyserlingii*) in the larval stage even 18 months after they completed the control group [32].

In natural populations of some salamandrid species, some individuals fail to metamorphose and retain larval traits into the adult stage. This phenomenon is called facultative pedomorphosis and occurs in several salamandrid species worldwide (Ambystomatidae, Plethodontidae, Salamandridae, Hynobiidae, and Dicamptodontidae) [33,34]. In two European newt species, *Ichthyosaura alpestris* and *Lissotriton vulgaris*, facultative pedomorphosis occurs in high incidence [35], but in crested newts of the genus *Triturus*, facultative paedomorphic individuals appear to be rare [33]. According to Wakahara (1996) [36], pedomorphosis could be explained either by deficiency in the production of THs or by a failure of the thyroid gland to become active and release its hormones into the bloodstream. Facultative pedomorphosis has an adaptive significance in newts and salamanders, allowing for either early reproduction or optimal use of resources and adaptation to local environmental conditions [33,37]. By retaining larval characteristics, animals avoid the cost of metamorphosis, whereas metamorphosed individuals can survive unfavorable conditions such as drying by dispersing to terrestrial habitats [38].

Since amphibian metamorphosis represents a unique example of TH-regulated development, it is of major interest to investigate how larvae adjust their physiological traits to reduced TH levels. Oxidative stress parameters; enzymes, including superoxide dismutase (SOD), catalase (CAT), glutathione peroxidase (GSH-Px), glutathione reductase (GR) and glutathione-S-transferase (GST); and nonenzymatic antioxidants, including glutathione (GSH), as well as thiobarbituric acid reactive substances (TBARS) and sulfhydryl (SH) groups, are considered as potential biomarkers that have been used as rapid screening tools in the assessment of different stress impacts on amphibians [39,40]. Thiourea treatment and altered thyroid function can be associated with oxidative stress indirectly through its action on the metabolism and development, and directly through effects on antioxidant mechanisms. Despite the known effect of thiourea on thyroid gland function and amphibian development, examination of oxidative stress parameters presents a new approach to the problems of altered TH syntheses in urodeles. The aim of this study was to estimate oxidative stress parameters in crested newt larvae (*Triturus* spp.) exposed to low (0.05%) and high (0.1%) thiourea concentrations and consequently arrested in the late larval stage of development.

## 2. Materials and Methods

### 2.1. Experimental Design

Larvae used in this study were reciprocal hybrids obtained from crosses between *Triturus ivanbureschi* and *Triturus macedonicus*. Experimental crossings were performed in March 2019 in common containers (large, 500 L containers were filled with water, closed with a protective net that contained plastic strips as underwater vegetation for egg deposition, bricks for shelter, and plastic floating islets). The collection of adult animals from natural populations was approved by the Ministry of Energy, Development and Environmental Protection of the Republic of Serbia (permit no. 353-01-75/2014-08) and the Environmental Protection Agency of Montenegro (permit no. UPI-328/4). The experiment was approved by the Ethical Committee of the Institute for Biological Research “Siniša Stanković” (decision no. 02-07/19). All experimental animals were treated in compliance with the European Directive (2010/63/EU) on the protection of animals used for experimental and other scientific purposes.

When females started depositing eggs, they were transferred to separate aquaria to lay their eggs. Eggs were collected daily and kept submerged in dechlorinated tap water in plastic Petri dishes until hatching, after which the hatchlings were transferred to a small plastic cup with dechlorinated tap water where they developed under laboratory conditions with a natural day-night light regime (12:12 h light-dark cycles) and water temperature of 16 ± 1 °C. At the early stages of development individuals were fed ad libitum with *Artemia* sp., and later with *Tubifex*.

The experiment was initiated at stage 62 and lasted throughout the late larval period until the larvae from the control group completed metamorphosis. The duration of exposure to thiourea lasted for about three months. The stages of development were determined according to the developmental staging table for newts [41] based on clearly visible morphological features (limb development and digit formation). Developmental stage 62 is characterized by fully developed limbs (formation of the fifth digit on the hindlimb), external gills and larval skin pigmentation. The first group of 14 animals was killed immediately at the outset of stage 62 (late larval stage). Another 42 animals were randomly assigned to three experimental groups as follows: (i) undisturbed development (dechlorinated tap water, control group—10 individuals); (ii) low thiourea concentration (0.05% solution of thiourea—16 individuals); (iii) high thiourea concentration (0.1% solution of thiourea—16 individuals).

Thiourea (p.a. ≥ 99.0%) was obtained from Sigma, St. Louis, MO, USA. Thiourea solutions (low and high concentrations) were prepared by dissolving thiourea in dechlorinated tap water. Two concentrations of thiourea (0.05% and 0.1%) were chosen according to previous work on salamanders [42]. 

To minimize confounding effects, larvae of the control and two treatment groups were kept under the same laboratory conditions at the same density until the end of metamorphosis of the control group. Two larvae per 2 L plastic containers half-filled with media were kept in order to avoid potential high larval density, which invariably led to a number of detrimental effects, such as reduction in larval growth or size at metamorphosis [43]. Because thiourea inhibits THs synthesis and therefore inhibits the ability of larvae to metamorphose, the end of the experiment was determined based on normally metamorphosing individuals in the control group. Metamorphosed individuals are characterized by the resorption of external gills and the closure of gill slits. Differences in the skin, especially in pigmentation and thickness, were prominent.

Before placing the individuals in liquid nitrogen, the body mass (BM) and body size (snout to vent length or SVL) were measured. Larvae were photographed using a Sony DSC-F828 digital camera (24-bit color and 3264 × 2448 pixel resolution, MP; Sony Corp., Tokyo, Japan). The SVL was measured using the TMorphGen6 program from the IMP package from photographs of the dorsal view of animals (from the tip of the snout to the level of the posterior edge of the cloaca). The body condition index (CI) was calculated from the log-transformed data for BM and SVL according to Labocha et al. (2014) [44].

### 2.2. Tissue Processing 

Whole larvae were chopped and mixed to obtain a homogenous material. One part of the tissue was used for analysis of the antioxidant parameters, and the remainder was used to measure the concentration of thiobarbituric acid reactive substances (TBARS) as a marker of lipid peroxidation. To measure antioxidant parameters, tissue homogenates were prepared in ice-cold 25 mM sucrose containing 10 mM Tris-HCl buffer (pH 7.5, at a ratio of 1:5) [45] using an Ultra-Turrax homogenizer (Janke & Kunkel, IKA-Werk, Staufen, Germany) [46]. Sonication was carried out at 20 kHz (30 s) on ice using an ultrasonic homogenizer (Sonopuls HD 2070, Bandelin electronic, Germany). A portion of each sonicate was centrifuged at 5000× *g* for 10 min in 10% sulfosalicylic acid and the resulting supernatants were used for the determination of GSH concentrations. Another portion was centrifuged at 100,000× *g* for 90 min at 4 °C and the obtained supernatants were used for the determination of other biochemical parameters. For TBARS estimation, tissues were suspended in ice-cold Tris-HCl homogenization buffer (pH 7.4) without sucrose in the 1:10 ratio and were then homogenized and sonicated. The suspensions were centrifuged at 10,000× *g* for 10 min at 4 °C in 40% trichloroacetic acid, and the obtained supernatants were used for the determination of TBARS.

### 2.3. Biochemical Analyses

The activity of SOD was assayed by the method of Misra and Fridovich [47], which is based on the ability of SOD to inhibit the auto-oxidation of epinephrine to adrenochrome at alkaline pH. The increase in absorbance was measured at 480 nm. CAT activity was determined by following the decrease in absorbance at 240 nm caused by the decomposition of hydrogen peroxide by CAT [48]. GSH-Px activity was analyzed as previously described [49]. Reaction was based on the oxidation of nicotinamide adenine dinucleotide phosphate (NADPH) with t-butyl hydroperoxide. The disappearance of NADPH was monitored at 340 nm. GR activity was determined by measuring NADPH oxidation at 340 nm during the reduction of oxidized glutathione (GSSG) [50]. The activity of the biotransformation II phase enzyme GST was determined spectrophotometrically at 340 by following the formation of the conjugate between 1-chloro-2,4-dinitrobenzene (CDNB) and GSH [51]. All enzyme activities are expressed as the specific activity (U/mg protein). Protein concentration was estimated by the method of Lowry et al. (1951) [52] using bovine serum albumin as standard. 

Quantification of the total concentration of GSH was based on the enzymatic recycling reaction in which GSH was oxidized by 5,5′-dithiobis (2-nitrobenzoic acid) (DTNB) and reduced by NADPH in the presence of GR [53]. The concentration of GSH was measured at 412 nm and expressed as nmol/g of tissue. The concentrations of free protein thiol (SH) groups were determined by Ellman’s method [54] expressed as μmol/g tissue. The reaction is based on the formation of a colored thiolate ion complex that was detected spectrophotometrically at 412 nm. The degree of lipid peroxidation was assessed by estimating the concentration of TBARS at 532 nm and is expressed in nmol/mg of tissues [55].

All measurements were performed using a UV-1800 UV-VIS spectrophotometer (Shimadzu, Kyoto, Japan) with a temperature-controlled cuvette holder. All chemicals were obtained from Sigma (St. Louis, MO, USA).

### 2.4. Statistical Analysis 

Outliers of the obtained data were identified by Grubb’s test, while the Kolmogorov-Smirnov test was used for testing normality. Levene’s test was used to check homogeneity of variance. Statistical differences in parameters between the two thiourea treatments (0.05% and 0.1% solutions) and the control groups (larvae stage 62 and metamorphosis) were analyzed via one-way analysis of variance (ANOVA) followed by Tukey’s HSD post-hoc test. Results are given as the mean ± SE. Canonical discriminant analysis was used to evaluate the differences among the investigated groups based on the measured parameters (SOD, CAT, GSH-Px, GR, GST, GSH, SH groups and TBARS). Standardized canonical discriminant function coefficients compared the relative importance of the independent variables (oxidative stress parameters) [56]. A value of *p* < 0.05 was considered as significant. STATISTICA 8.0 software (StatSoft, Tulsa, OK, USA) was used for statistical analyses.

## 3. Results

Values for SVL, BM, and CI are presented in Table 1. All biometric parameters were significantly lower in the late larval stage in comparison with the other three experimental groups.

The results of oxidative stress parameters are presented in Figure 1. For all investigated parameters, the results of the ANOVA analysis are presented in Table 2 as F values and the level of statistical significance (*p*). SOD activity was significantly higher in newly metamorphosed individuals (7.41 ± 0.37) as compared to individuals in the late larval period (5.63 ± 0.33) and those exposed to 0.05% thiourea solution (4.88 ± 0.23). SOD activity was also higher in individuals that were exposed to the 0.1% thiourea solution (7.30 ± 0.25) in comparison to the late larval stage and larvae exposed to the low (0.05%) thiourea concentration. CAT activity was higher in newly metamorphosed crested newts (44.48 ± 3.10) in comparison to all other examined groups (late larval period (27.78 ± 1.68), larvae exposed to 0.05% thiourea concentration (20.29 ± 1.38), and larvae exposed to 0.1% thiourea concentration (16.72 ± 0.95). CAT was also higher in the late larval stage in comparison to both thiourea treatments. Our results showed the same pattern of change for GSH-Px and GR activities. The newly metamorphosed individuals had significantly higher activities of these enzymes (11.54 ± 1.22 and 12.88 ± 0.85 for GSH-Px and GR, respectively) compared to the late larval period (6.96 ± 0.37 and 8.44 ± 0.56 for GSH-Px and GR, respectively) and individuals that were exposed to the 0.1% thiourea solution (6.38 ± 0.44 and 9.19 ± 0.37 for GSH-Px and GR, respectively). At the same time, the activities of GSH-Px (10.42 ± 0.50) and GR (12.22 ± 0.55) were significantly higher in individuals reared in a low (0.05%) thiourea solution relative to the late larval and the group exposed to a high (0.1%) thiourea concentration. The activity of GST and the concentrations of SH groups exhibited the same mode of change. The highest values of these parameters (851.49 ± 57.08 and 1370.32 ± 42.60 for GST and SH groups, respectively) were recorded in the newly metamorphosed individuals, while the lowest values (189.62 ± 9.57 and 605.03 ± 13.47 for GST and SH groups, respectively) were observed in the late larval stage. The GSH content was significantly higher in the late larval stage (312.17 ± 11.03) in comparison to larvae exposed to 0.1% thiourea solution (252.65 ± 8.61) (Figure 1). The concentration of TBARS differed significantly between all examined groups. The highest value was measured in the late larval stage (9.51 ± 0.43), followed by the group exposed to the 0.1% thiourea treatment (8.00 ± 0.31), and newly metamorphosed (6.09 ± 0.36), and the lowest value was obtained for individuals exposed to the 0.05% thiourea solution (4.61 ± 0.27).

Canonical discriminant analysis of oxidative stress parameters is presented in Figure 2. Untreated larvae in ontogenetic stage 62 were separated from the other three investigated groups when considering their relative position in the first canonical function (Root 1). Root 2 showed clear separation between larvae treated with a high (0.1%) thiourea concentration and larvae that were treated with a low (0.05%) thiourea concentration, as well as metamorphosed individuals. Standardized coefficients for canonical variables (Table 3) showed that SH groups (−0.778), CAT (0.718), and TBARS (0.620) mostly influenced differences in Root 1, while SOD (−0.957) and CAT (0.766) influenced differences in Root 2.

## 4. Discussion

Urodele evolution is accompanied by the concentration of ontogenetic events within a metamorphic period and the progressive increase of the regulatory role of THs. In primitive salamanders (fam. Hynobiidae), many adult specializations develop as early as in larvae, and developmental events are activated by environmental factors and inductive tissue interactions, while in advanced species (fam. Salamandridae and Plethodontidae) their development is shifted toward metamorphosis and is hormonally regulated [5]. 

During metamorphosis, the entire organism changes to prepare itself for its new mode of existence. Accumulated developmental events and organ remodeling require physiological and molecular adaptations that are crucial to enable optimal survival of individuals [57,58]. In amphibians it was suggested that oxidative stress is important in the regulation of organ resorption and remodeling during metamorphosis. During transition to the terrestrial environment, which is often richer in oxygen, an adequate response to increased endogenous production of reactive oxygen species is crucial in preventing increased oxidative injuries [59]. Our previous study showed that *Pelophylax esculentus* tadpoles in metamorphic climax displayed higher levels of TBARS and lower activities of SOD, CAT, and GR than individuals that have completed metamorphosis [60]. Oxidative stress resulting from a decrease in CAT and GR could be one of the mechanisms underlying cell death during intestinal remodeling and tail regression in anurans [57]. Since adult urodeles retain their tails, it is clear that some of the most obvious external features of anuran metamorphosis will not be observed and the metamorphic process is not as dramatic as metamorphosis in anurans [3]. Nonetheless, the adjustment of their physiological responses can be important for understanding the challenge of metamorphosis and the transition to the terrestrial environment. Our results show that larvae at the beginning of developmental stage 62, a stage followed by intensive growth and complete limb formation, were characterized by a lower antioxidant response and highest lipid oxidative damage. This indicated that individuals in this developmental stage were more susceptible to oxidative stress. In contrast, increased activities of the antioxidant enzyme machinery seen in newly metamorphosed individuals could allow them to adopt and conquer new terrestrial habitats and switch to the terrestrial mode of life [61,62]. A dramatic change in GST expression was noted in animals that left the aquatic habitats after metamorphosis to live mostly in the terrestrial environment, while analogous changes were not observed in species that never leave the aquatic ecosystem [63,64,65]. Our results showed that there was a significant increase in GST activity in metamorphosed individuals compared to those in the late larval stage. Based on the results obtained by canonical discriminant analysis, TBARS concentration, SH content, and CAT activity contributed the most to the separation of larvae compared to metamorphosed individuals and individuals whose development was artificially arrested.

Even at very low concentrations (30 ppm), thiourea, as a potent endocrine disruptor, can slow down metamorphosis in poikilothermic animals [66]. The THs of crested newt larvae reared in both aqueous solutions containing thiourea were inhibited. Animals did not metamorphose but continued to grow while retaining their larval form. Impairment of the hypothalamus-pituitary-thyroid axis affects metabolic costs and energy allocation to such an extent that thyroidectomized tadpoles can reach almost twice the length of normal larvae [30,67,68]. These changes in metabolic rate and energy allocation are closely coupled to the generation of reactive oxygen species and lead to modulation of the antioxidant defense system. Although the metabolic effect of THs is associated with the acceleration of basal metabolism, both hyperthyroid and hypothyroid conditions can change the oxidant-antioxidant state in tissues [69]. Under these circumstances, the marked upregulation of antioxidant enzyme mRNA expression helps the cell to maintain the redox potential. Bhanja and Chainy (2010) [70] reported that 6-n-propylthiouracil-induced hypothyroidism alters the expression of antioxidant enzymes and triggers an increase in H_2_O_2_ in rat tissues. Therefore, increased expression of enzymatic antioxidant components, especially of CAT, GSH-Px, and SOD, could shield tissues from the toxic effect of H_2_O_2_ and thereby prevent damage of cellular components. On the other hand, antioxidative enzymes could be targets for antithyroid drugs and therefore their inhibition might be relevant in inducing cellular dysfunction and toxicity [71]. We found significant reductions in SOD, CAT and GST enzyme activities in crested newts reared in the 0.05% aqueous thiourea solution as compared to individuals that completed metamorphosis. GSH-Px and GR activities, though non-significantly, were also lower in treated animals. These animals also had the lowest concentration of TBARS, pointing to the positive effects of thiourea on the oxidative status of treated larva. The higher (0.1%) thiourea concentration had a more intense inhibitory effect on the activity of all enzymes except SOD. The activity of SOD in larvae reared in an aqueous solution containing a higher concentration of thiourea was significantly induced compared to larvae reared in a solution with a lower thiourea concentration, reaching the level of activity similar to that measured in metamorphosed individuals. We hypothesized that the induction of SOD enzyme activity in larvae treated with 0.1% thiourea may be related to increased superoxide anion radical production and the toxic effect of a higher thiourea concentration. SOD is the principal enzyme for the elimination of superoxide radicals produced during metabolism. It catalyzes its dismutation to hydrogen peroxide and oxygen, while CAT and GSH-Px react with accumulated hydrogen peroxide [72]. The lowest GSH content was measured in newt larvae reared in high thiourea concentration and was not sufficient to prevent oxidative damage resulting from thiourea toxicity. Thiourea toxicity was also shown to be accompanied by a depletion of intracellular GSH in rat hepatocytes [73]. Overall, the response of all antioxidative defense parameters in crested newt larvae exposed to higher thiourea concentrations was not capable of preventing oxidative stress.

Our results revealed that crested newts respond to thiourea exposure with interrupted development, as in other amphibians [30,69,74]. A significant increase in body size relative to the duration of the experiment was not observed since we did not monitor the effects on thiourea-treated individuals in the period after metamorphosis. In our study, the CI showed that thiourea had no negative effect on the general health of crested newts whose metamorphosis was stopped. In the so-called synchronized amphibian metamorphosis assay, a high concentration of thiourea solution (0.2%) was used to obtain tadpoles available at any time for laboratory metamorphosis experiments, and to acquire experimental groups that were much more homogeneous [75]. In some circumstances there is a need to maintain salamanders in the larval stage because the feeding of metamorphosed individuals is not possible in laboratory conditions, and therefore researchers resort to an artificial delay of metamorphosis [76]. Our results warn of the physiological effect of thiourea which must not be ignored, especially if thiourea could be used as a pretreatment to synchronize development or to delay metamorphosis in toxicological studies.

## 5. Conclusions

This study revealed that the responses of enzymatic antioxidant components (SOD, CAT, GSH-Px, GR, and GST) in the course of undisturbed development of crested newts were sufficient to prevent peroxidative damage during intense growth and energy allocation, making newly metamorphosed individuals ready for the transition to a new form of existence and to spread to terrestrial habitats. The synthesis of THs of crested newt larvae that were reared in aqueous thiourea solutions were artificially inhibited: They did not metamorphose but continued to grow, retaining larval characteristics. It seems that the lower thiourea concentration exerted protection as it reduced the TBARS level, while at high concentrations, thiourea disturbed the oxidative balance and modulated physiological parameters to the extent that oxidative lipid damage could not be avoided. Examination of the physiological alterations in arrested metamorphosis is significant because it can reveal adaptive patterns in pedomorphic newts that in natural conditions delay metamorphosis, but whose examination is difficult considering that they are a protected and endangered species.

## Figures and Tables

**Figure 1 ijerph-18-09653-f001:**
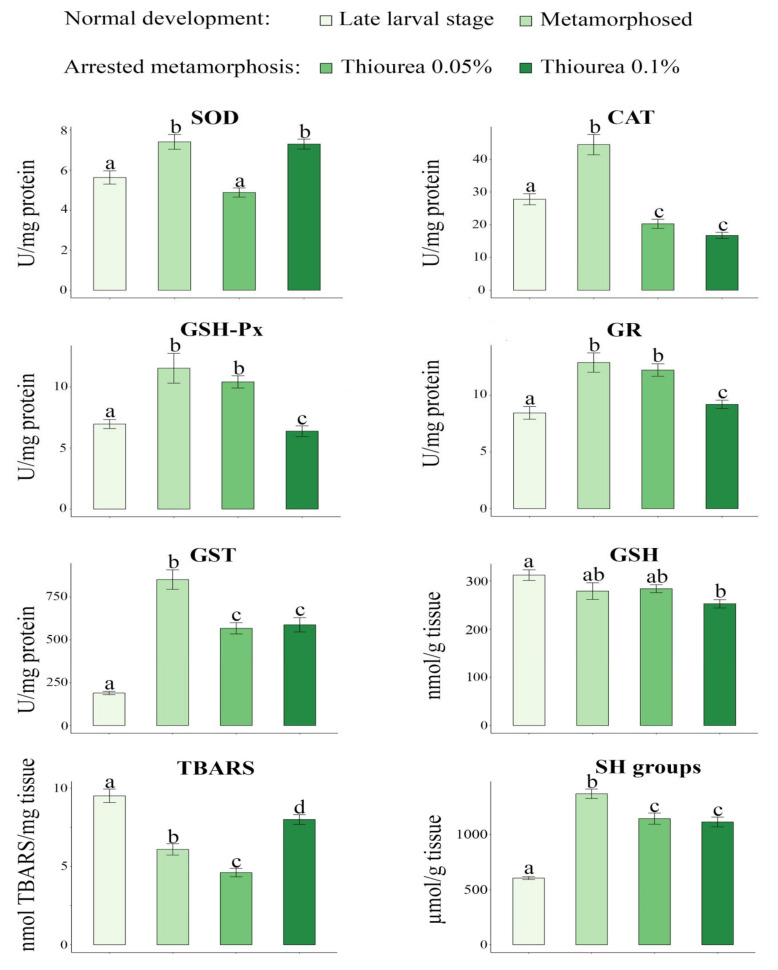
Oxidative stress parameters: superoxide dismutase (SOD), catalase (CAT), glutathione peroxidase (GSH-Px), glutathione reductase (GR), glutathione-S-transferase (GST), glutathione (GSH), thiobarbituric acid reactive substances (TBARS), and sulfhydryl (SH) groups of crested newts (*Triturus* spp.) in late larval stage (62 stage) (n = 14), newly metamorphosed (n = 10), and metamorphosis-arrested (exposed to low (0.05%) (n = 16) and high (0.1%) thiourea concentrations) (n = 16) individuals. The data are expressed as the mean ± SE. The letters ^a^, ^b^, ^c^ and ^d^ show the results of statistical analysis (Tukey’s HSD, *p* < 0.05). n—number of individuals per group. The same letter indicated that the difference between the groups was not significant. Different letters indicated that they were significantly different.

**Figure 2 ijerph-18-09653-f002:**
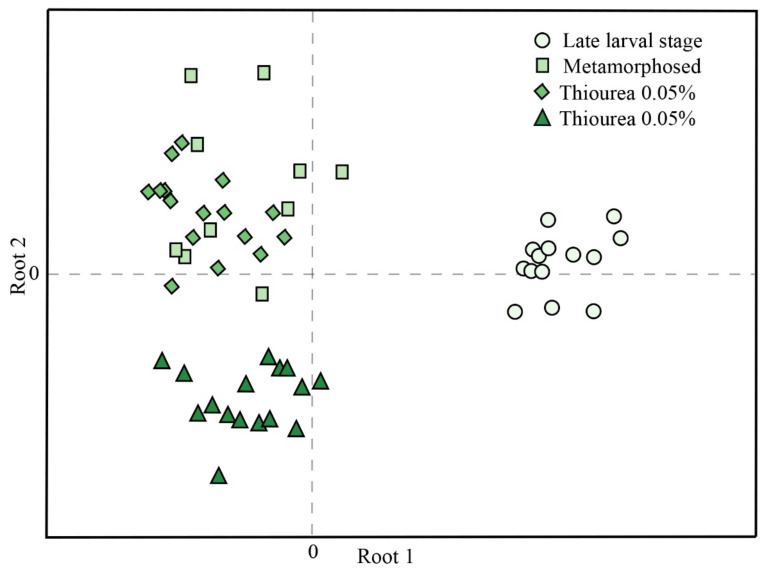
Canonical discriminant analysis of oxidative stress parameters (SOD, CAT, GSH-Px, GR, GST, GSH, TBARS, and SH groups) in crested newts (*Triturus* spp.) in late larval stage (62 stage) (n = 14), newly metamorphosed (n = 10), and metamorphosis-arrested (exposed to low (0.05%) (n = 16) and high (0.1%) (n = 16) thiourea concentrations) individuals. Groups were formed by Root 1 (the first canonical function) and Root 2 (the second canonical function). n—number of individuals per group.

**Table 1 ijerph-18-09653-t001:** The total snout-vent length (SVL), body mass (BM), and body condition index (CI) of late larval stage (62 stage), newly metamorphosed, and metamorphosis-arrested (after exposure to low (0.05%) and high (0.1%) thiourea concentrations) crested newts. SVL and BM are given as the mean ± SE, the CI was calculated on log-transformed data. The letters ^a^ and ^b^ show the results of statistical analysis (Tukey HSD, *p* < 0.05). The same letter indicated that the difference between the groups was not significant. Different letters indicated that they were significantly different.

	Late Larval Stage	Newly Metamorphosed	Thiourea 0.05%	Thiourea 0.1%
SVL (mm)	20.69 ± 0.38 ^a^	34.47 ± 0.80 ^b^	33.56 ± 0.69 ^b^	32.85 ± 0.75 ^b^
BM (g)	0.45 ± 0.03 ^a^	1.69 ± 0.10 ^b^	1.93 ± 0.12 ^b^	1.81 ± 0.11 ^b^
CI	0.12 ± 0.01 ^a^	0.28 ± 0.01 ^b^	0.30 ± 0.01 ^b^	0.29 ± 0.01 ^b^

**Table 2 ijerph-18-09653-t002:** Results of one-way ANOVA of the comparison between late larval stage (62 stage), newly metamorphosed, and metamorphosis-arrested (after exposure to low (0.05%) and high (0.1%) thiourea concentrations) crested newts.

Variable	F	*p*
CI	79.99	0.000000
SOD	18.89	0.000000
CAT	43.98	0.000000
GSH-Px	16.56	0.000000
GR	14.34	0.000001
GST	50.14	0.000000
GSH	5.60	0.002025
TBARS	38.36	0.000000
SH groups	55.44	0.000000

**Table 3 ijerph-18-09653-t003:** Standardized coefficient for canonical variables (oxidative stress parameters).

Variable	Root 1	Root 2
SH groups	−0.778	−0.310
CAT	0.718	0.766
TBARS	0.620	−0.198
GST	−0.494	0.202
GSH	0.392	0.081
SOD	−0.222	−0.957
GR	−0.134	0.192
GSH-Px	0.014	0.439

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
