# Peer review of "Oxidative Stress Parameters in Goitrogen-Exposed Crested Newt Larvae (Triturus spp.): Arrested Metamorphosis"

_ijerph, 2021, doi:10.3390/ijerph18189653_

Round 1

Reviewer 1 Report

The authors of the manuscript entitled “Oxidative stress parameters in goitrogen-exposed crested newt larvae (Triturus spp.): arrested metamorphosis investigated the oxidative stress parameters in crested newts at different stages of development, larval, newly metamorphosed, and arrested metamorphosis. The authors found that the synthesis of THs in crested newt larvae reared in aqueous thiourea solutions was inhibited, but continued to grow, retaining larval characteristics. The authors concluded that lower thiourea concentration exerted protection as it reduced the TBARS level, while high concentrations of thiourea disturbed the oxidative balance. The study is novel and has the potential to be accepted for publication provided the authors can address the critical issues listed below.

  1. The topic contradicts the objective of the study. The author investigated the oxidative parameters of crested newt larvae (Triturus spp) at different stages of development. Thiourea was only used to artificially induce arrested metamorphosis condition, in this case, it is not a treatment. Therefore, the topic should be changed
  2. The English with which this manuscript was written are below standard, the manuscript from the abstract is difficult to follow. Kindly fix syntax problems.
  3. The method and result are not well detailed in the abstract. The abstract is not a good representative of the study. Kindly redraft
  4. Line 26, all abbreviations must be written in full at first mention
  5. Since the authors used thiourea to induce a developmental condition (arrested metamorphosis), authors should have optimized the concentration of thiourea that best arrest the metamorphosis. As it is now, the statistical analysis used is not appropriate because there are two factors- developmental stages and thiourea concentrations. Authors could report 0.1 % thiourea as arrested metamorphosis.
  6. The superscript assignment in figures 1c and 1d is confusing, clarify the difference between a, b, and c
  7. The authors should write comprehensively on thiourea as an endocrine-disrupting chemical

Author Response

Reviewer #1

General comment:

The authors of the manuscript entitled “Oxidative stress parameters in goitrogen-exposed crested newt larvae (Triturus spp.): arrested metamorphosis investigated the oxidative stress parameters in crested newts at different stages of development, larval, newly metamorphosed, and arrested metamorphosis. The authors found that the synthesis of THs in crested newt larvae reared in aqueous thiourea solutions was inhibited, but continued to grow, retaining larval characteristics. The authors concluded that lower thiourea concentration exerted protection as it reduced the TBARS level, while high concentrations of thiourea disturbed the oxidative balance. The study is novel and has the potential to be accepted for publication provided the authors can address the critical issues listed below.

We are grateful for the insightful comments that improved our paper. We have incorporated most of your suggestions. Changes made according to your suggestions are highlighted with yellow color in the main text.

Specific comments:

  1. The topic contradicts the objective of the study. The author investigated the oxidative parameters of crested newt larvae (Triturus spp) at different stages of development. Thiourea was only used to artificially induce arrested metamorphosis condition, in this case, it is not a treatment. Therefore, the topic should be changed.

Thank you for pointing this out. In the revised version of manuscript we aligned the topic with the objectives of the study and made changes in the text, especially in the abstract. Using the statement "during different stages of metamorphosis" we may have lead to inadequate interpretation of the study topic. We investigated the oxidative stress parameters of crested newt larvae during normal development and thiourea treatment, which due to its goitrogenic effect, induced arrested metamorphosis. Thiourea has been used as a treatment in many previous studies (Gordon et al. (1943); Joel et al. (1949), Krishnapriya et al. (2014)). In our study, thiourea exposure led to metamorphosis arrest, and this could be associated with oxidative stress not only indirectly through its action on the development, but also through its direct effects on antioxidant parameters. We mentioned this in the lines 36-40.

  1. The English with which this manuscript was written are below standard, the manuscript from the abstract is difficult to follow. Kindly fix syntax problems.

The present version was again submitted for proofreading. We hope that the English level now reaches the expected quality.

  1. The method and result are not well detailed in the abstract. The abstract is not a good representative of the study. Kindly redraft

We changed the abstract according to the suggestion.

  1. Line 26, all abbreviations must be written in full at first mention

We made the correction. The abbreviation was defined.

  1. Since the authors used thiourea to induce a developmental condition (arrested metamorphosis), authors should have optimized the concentration of thiourea that best arrest the metamorphosis. As it is now, the statistical analysis used is not appropriate because there are two factors- developmental stages and thiourea concentrations. Authors could report 0.1 % thiourea as arrested metamorphosis.

Two concentrations of thiourea (0,05% and 0,1%) were chosen according to previous work on salamanders (Wheeler, 1953). Author described effects of various concentration of thiourea (0.005%, 0.01%, 0.05%, 0.1%, 0.3%, 0.5% and 1.0%) on histology of thyroid gland of salamanders. The results of this study showed that 0.05% and 0.1% solutions were the most optimal concentrations of thiourea that caused goitrogenic effects in salamanders. Our results confirmed that both group of thiourea-treated individuals retainin their larval characteristics until the end of the experiment and they did not complete metamorphosis.

We started the experiment when the larvae reached stage 62(differentiation of forelimbs, posterior limbs and toes were finished). This stage was selected because it is the longest and final stage of larval development, during which individuals grow without any greater morphological changes. The first group of animals was killed immediately at the outset of stage 62 (Late larval stage), representing first control group. The second group (newly metamorphosed) was control individuals reared until the metamorphosis. The third and fourth groups of animals (Thiourea 0.05% and Thiourea 0.1% metamorphosis-arrested individuals) were reared in thiourea solutions, 0.05% and 0.1% respectively. When control individuals metamorphosed we finished the experiment. As the late larval, Thiourea 0.05% and Thiourea 0.1% groups represented larvae at the same stage of development – stage 62, we consider that the developmental stages cannot be an independent factor.

  1. The superscript assignment in figures 1c and 1d is confusing, clarify the difference between a, b, and c

We deleted superscripts in Figure 1. For groups with the same letters the difference was not significant. Likewise, for groups with different letters, the differences were statistically significant. Clarification can be found in the text: "Different letters indicate significant differences between groups (Tukey’s HSD, p<0.05)." (line 421).

  1. The authors should write comprehensively on thiourea as an endocrine-disrupting chemical.

We agree with your suggestion. The endocrine disrupting effect of thiourea was discussed (lines 85-93):

"Studies have shown that thiourea is an endocrine-disrupting chemical that can alter the thyroid function of different organisms (fish, amphibians, lizards, mammals) [20,24,25]. It was shown that thiourea acts via inhibition of thyroid peroxidase, an enzyme that is indispensable to THs synthesis [26,27]. The endocrine-modulating properties of thiourea were evidenced by depression of the levels of THs, a lower metabolic rate, as well as other symptoms of hypothyroidism. In thiourea-treated fish, the level of T4 was unchanged; however, T3 was significantly lower than in the control group, which pointed to the effect of thiourea on the rate of T4 deiodination [28]."

References

  1. Wheeler, A.J. Temporal variations in histological appearance of thyroid and pituitary of Salamanders treated with thyroid inhibitors. Bull. 1953, 104(2), 250–262.

Reviewer 2 Report

the work of Dr. Jelena P. Gavrić and colleagues is well done and presented.
I suggest the following

1. improve the quality and presentation of figure 1.
2 it is confusing to put letters to each figure (a, b, c, d, f, g, h) and to each group (a, b, c, d) of figure 1. or is it the statistics? figure caption what does each letter mean? Thank you
3. When thiourea is administered at a high dose (0.1%), TBARS rises, and antioxidant enzymes decrease, only SOD rises, what does it mean?
4. In table 3, there is numbering, I think it is the numbering of the lines, please check
5. I suggest that in the discussion some regulation mechanism of antioxidant enzymes be postulated in this model. Thank you

Author Response

Reviewer #2

General comment:

The work of Dr. Jelena P. Gavrić and colleagues is well done and presented.

We are very thankful for your comments on our work. We appreciate the time and effort that you dedicated to providing feedback on our manuscript. Changes made according to your suggestions are highlighted with turquoise in the main text.

Specific comments:

  1. improve the quality and presentation of figure 1.

We improved the quality of both figures.

  1. it is confusing to put letters to each figure (a, b, c, d, f, g, h) and to each group (a, b, c, d) of figure 1. or is it the statistics? figure caption what does each letter mean? Thank you

We deleted superscripts in Figure 1. The same letter indicated that the difference between the groups was not significant. Different letters indicated that they were significantly different. Clarification can be found in the text: "Different letters indicate significant differences between groups (Tukey’s HSD, p<0.05)." (line 421).

  1. When thiourea is administered at a high dose (0.1%), TBARS rises, and antioxidant enzymes decrease, only SOD rises, what does it mean?

SOD is a principal enzyme for the elimination of superoxide radicals and we hypothesized that the induction of SOD enzyme activity in larvae treated with 0.1% thiourea may be related to the toxic effect of higher thiourea concentration (lines 542-546).

  1. In table 3, there is numbering, I think it is the numbering of the lines, please check

We checked the Table 3.

  1. I suggest that in the discussion some regulation mechanism of antioxidant enzymes be postulated in this model. Thank you

Thank you for a very useful suggestion. In the revised manuscript version, we have discussed the some regulation mechanism of antioxidant enzymes. (lines 522 - 531):

"Although the metabolic effect of THs is associated with the acceleration of basal metabolism, both hyperthyroid and hypothyroid conditions can change the oxidant-antioxidant state in tissues [69]. Under these circumstances, the marked upregulation of antioxidant enzyme mRNA expression helps the cell to maintain the redox potential. Bhanja and Chainy (2010) [70] reported that 6-n-propylthiouracil-induced hypothyroidism alters the expression of antioxidant enzymes and triggers an increase in H2O2 in rat tissues. Therefore, increased expression of enzymatic antioxidant components, especially of CAT, GSH-Px and SOD, could shield tissues from the toxic effect of H2O2 and thereby prevent damage of cellular components."

References

  1. Oktay, S.; Uslu, L.; Emekli, N. Effects of altered thyroid states on oxidative stress parameters in rats. Basic. Clin. Physiol. Pharmacol. 2017, 28(2), 159–165.
  2. Bhanja, S.; Chainy, G.B.N. PTU-induced hypothyroidism modulates antioxidant defence status in the developing cerebellum. J. Dev. Neurosci. 2010, 28(3), 251–262.

Reviewer 3 Report

The article by Gavric and colleagues is well written and can be published in IJERPH after some modifications.  

Specific comments:

Line 110: Please write full scientific names when starting a new section (e.g. M&Ms)

Lines 116-119: Is the facility licensed according to the European Directive 2010/63/EU and the national legislation? Was the experimental protocol licensed by the national authorities for experimental animals?

Line 125: Please define the natural day-night light regime.

Lines 208-209: Were the data tested for equality of variances? Please define.

Figures: Please note sample size (n) in each graph because the sample number is not quite clear

In the Results section, please write down the actual measurement when applicable. For example, “CAT activity was higher in newly metamorphosed crested newts (MEAN±SD) in comparison to all other examined groups.

Table 2. I really cannot figure out the significance of the table. These F values are not the same as in Figure 1?

Author Response

Reviewer #3

General comment:

The article by Gavric and colleagues is well written and can be published in IJERPH after some modifications.

We are very glad that you liked our paper and are thankful for quality suggestions, and the time dedicated to helping us to improve the quality of our manuscript. Changes made according to your suggestions are highlighted with green in the main text.

Specific comments:

  1. Line 110: Please write full scientific names when starting a new section (e.g. M&Ms)

The full scientific names were added. (Line 136)

  1. Lines 116-119: Is the facility licensed according to the European Directive 2010/63/EU and the national legislation? Was the experimental protocol licensed by the national authorities for experimental animals?

All experimental animals were treated in compliance with the European Directive (2010/63/EU) on the protection of animals used for experimental and other scientific purposes (line xx-yy). Collection of adult animals from natural populations was approved by the Ministry of Energy, Development and Environmental Protection of the Republic of Serbia (permit no. 353-01-75/2014-08) and the Environmental Protection Agency of Montenegro (permit no. UPI-328/4). The experiment was approved by the Ethical Committee of the Institute for Biological Research “Siniša Stanković” (decision no. 02-07/19) (lines 139-146).

  1. Line 125: Please define the natural day-night light regime.

The natural day-night light regime was defined in line XX (12:12h light-dark cycles). (line 151)

  1. Lines 208-209: Were the data tested for equality of variances? Please define.

Levene's test was used to check the homogeneity of variance (lines 275-276).

  1. Figures: Please note sample size (n) in each graph because the sample number is not quite clear

We noted sample size (lines 419-422; 469-472)

  1. In the Results section, please write down the actual measurement when applicable. For example, “CAT activity was higher in newly metamorphosed crested newts (MEAN±SD) in comparison to all other examined groups.

According to your suggestion, we added values (mean±SE) for each examined parameter.

  1. Table 2. I really cannot figure out the significance of the table. These F values are not the same as in Figure 1?

F values were not presented in Figure 1. In this figure we presented values for each examined oxidative stress parameter, expressed as the mean±SE. Statistical difference between the investigated groups was analyzed via the one-way ANOVA which uses F-tests to statistically assess the equality of means. F and p values of this analysis were presented in Table 2. Since ANOVA results do not identify which particular differences between means of investigated groups are significant, we used Tukey HSD post hoc test to identify where the specific differences were. Results of post the hoc test were presented with lowercase letters (above the columns) in Figure 1. We made the correction in the text (line 421).

Round 2

Reviewer 1 Report

The authors have addressed all the issues raised during the earlier review